# Utility of Alternative Promoters for Foreign Gene Expression Using the Baculovirus Expression Vector System

**DOI:** 10.3390/v14122670

**Published:** 2022-11-29

**Authors:** Mark R. Bruder, Marc G. Aucoin

**Affiliations:** Department of Chemical Engineering, University of Waterloo, Waterloo, ON N2L 3G1, Canada

**Keywords:** promoters, baculovirus expression vector system, BEVS, virus-like particle, VLP, p6.9, 39k, vp39, ctx, orf75, 38k, gp64

## Abstract

The baculovirus expression vector system (BEVS) is a widely used platform for recombinant protein production for use in a wide variety of applications. Of particular interest is production of virus-like particles (VLPs), which consist of multiple viral proteins that self-assemble in strict stoichiometric ratios to mimic the structure of a virus but lacks its genetic material, while a significant amount of effort has been spent on optimizing expression ratios by co-infecting cells with multiple recombinant BEVs and modulating different process parameters, co-expressing multiple foreign genes from a single rBEV may offer more promise. However, there is currently a lack of promoters available with which to optimize co-expression of each foreign gene. To address this, previously published transcriptome data was used to identify promoters that have incrementally lower expression profiles and compared by expressing model cytoplasmic and secreted proteins. Bioinformatics was also used to identify sequence determinants that may be important for late gene transcription regulation, and translation initiation. The identified promoters and bioinformatics analyses may be useful for optimizing expression of foreign genes in the BEVS.

## 1. Introduction

The Baculovirus Expression Vector System (BEVS) is a versatile manufacturing platform for clinically important therapeutic proteins, viral vectors for gene therapy, and antigens for vaccination. In addition to being scalable, cost efficient, and capable of high product yields, the ability to rapidly produce a recombinant BEV (rBEV) carrying large foreign DNA inserts has catapulted the BEVS platform toward mainstream acceptance in the biotechnology industry [1]. The current COVID-19 pandemic has underscored the importance of the BEVS for manufacturing antigens for vaccination: the SARS-CoV-2 vaccine NVX-CoV2373 developed by Novavax and produced using the BEVS platform is just the 4th vaccine approved for Emergency Use Authorization (EUA) by the US Food and Drug Administration (FDA), and is the first protein subunit vaccine approved against COVID-19 [2].

Although the BEVS has been used as an expression vector for thousands of recombinant proteins, the vast majority of research has focused on expression of a single foreign gene using the very strong, very late, polh or p10 promoters [3,4]. Despite this, many studies have suggested that earlier promoters may be more effective for producing some foreign proteins, particularly those that require extensive post-translational processing and/or secretion, as the very late promoters are most active when host cell processes may be compromised [3,5,6,7,8]. For more complex biologics that are composed of multiple foreign proteins such as virus-like particles (VLPs), co-infection with multiple monocistronic baculoviruses is a strategy that is routinely employed [3]. Co-infection allows for optimization of process parameters such as multiplicity of infection (MOI) and time of infection (TOI) for each individual rBEV to modulate the timing of expression and stoichiometry of each protein to maximize self-assembly and overall yield. Substantial improvements in the production of adeno-associated virus (AAV) and rotavirus-like particles have been realized with this approach [3,9,10,11]. Co-infection with multiple monocistronic baculoviruses does have potential disadvantages, however: various studies have suggested that as the number of viruses increases, the proportion of cells that are infected with each virus (or in equal ratios) decreases, leading to efficiency loss in production of fully formed VLPs. Further, the burden of copying genetic material of multiple different viruses may lead to faster cell death [12]. Polycistronic baculoviruses, on the other hand, ensure that every protein necessary for the self-assembly of the VLP is expressed in each infected cell, and various studies have reported higher yields using co-expressed proteins from polycistronic baculoviruses than co-infections with multiple monocistronic vectors [3]. Due to the predominance of the p10 and polh promoters on commercially available transfer vectors, however, modulating expression parameters akin to MOI and TOI is virtually impossible.

Despite the aforementioned utility of promoters that are active earlier in the infection cycle for foreign gene expression, previous reports have focused mainly on identifying endogenous or chimeric baculovirus promoters that achieve higher protein expression levels than the native polh or p10 promoters [5,13,14,15,16]. For complex VLPs, however, individual proteins may be required in very different amounts; for example, the abundance of the four distinct proteins that make up rotavirus-like particles ranges from 60 to 780 molecules per particle, and the ratios of the integral membrane proteins hemagglutinin (HA), neuraminidase (NA), and matrix (M2) of influenza A virus is approximately 4:1:0.04-0.4 (HA:NA:M2) [9,17]. Moreover, for many non-enveloped VLPs such as rotavirus-like particles, a fixed stoichiometry between the four constituent proteins is required for properly assembled and stable particles. This presents a considerable challenge for co-expression of the required molecules from a polycistronic rBEV, as the expression levels of multiple proteins, each having unique characteristics, must be modulated to satisfy strict stoichiometric requirements. However, the catalogue of promoters that have been reported so far for the BEVS includes very few that have lower expression profiles than p10/polh.

To improve upon the catalogue of rBEV promoters, we used previously reported transcriptome data [18] to select a series of native *Ac*MNPV promoters with different expression characteristics. The promoters were evaluated and compared by expressing the model cytoplasmic protein green fluorescent protein (GFP) and secreted protein secreted alkaline phosphatase (SEAP). Finally, we aimed to evaluate whether sequence determinants were identifiable that govern the expression characteristics of late *Ac*MNPV gene expression. In addition to adding new promoters to the rBEV expression catalogue, to our knowledge this is the first report of employing bioinformatics analyses for evaluation of *Ac*MNPV promoters. It is our hope that this report may help improve production of complex biologics such as VLPs from polycistronic rBEVS by allowing for more modulation of expression of each foreign gene, as well as spur on more comprehensive evaluation of gene expression profiles to further our understanding of baculovirus gene expression and improve the BEVS as a biotechnology platform.

## 2. Materials and Methods

### 2.1. Plasmid Construction

All plasmids used in this study were constructed using the NEBuilder HiFi DNA Assembly Master Mix (New England Biolabs, Whitby, ON, Canada) according to manufacturer’s directions. Primers used for construction of all plasmids were synthesized by Integrated DNA Technologies (IDT; Coralville, IA, USA) and are given in Appendix A. The genomic regions for each promoter are given in Table 1. The co-ordinates are based on the RefSeq entry for the *Ac*MNPV genome (NC_001623.1) [19].

To construct the promoter-GFP transfer plasmids, the selected promoter regions were amplified from *Ac*MNPV genomic DNA and inserted upstream of the *gfp* gene encoding green fluorescent protein (GFP) [20] in plasmid p6.9-GFP, described previously [21]. For SEAP-expressing plasmids, the *gfp* gene in each of the promoter-GFP plasmids was replaced with the *seap* gene encoding secreted alkaline phosphatase (SEAP) amplified from pYSEAP (Addgene # 37326).

### 2.2. Recombinant Baculovirus Generation, Amplification, and Quantification

Transfer plasmids for recombinant baculovirus expression vector (rBEV) generation were co-transfected with *flash*BACGOLD™ (Oxford Expression Technologies Ltd., Oxford, UK) genomic DNA to Sf9 cells using Escort IV transfection reagent (Sigma-Aldrich, Oakville, ON, Canada) according to manufacturer’s directions. Supernatant from each transfection was harvested 4–5 days post transfection and used to infect suspension Sf9 cultures (∼$1.5\times 10^{6}$ cells/mL) at low multiplicity of infection (MOI) for 3–4 days to amplify the rBEV. Following two rounds of amplification, the rBEV infectious virus titer (IVT) was quantified using end-point dilution assay (EPDA). Briefly, Sf9 cells were seeded at a density of ∼$2.0\times 10^{4}$ cells/well to each well of a 96-well plate (Fisher Scientific, Whitby, ON, Canada). Separately, the rBEV was serially diluted ($10^{-2}$ to $10^{-8}$) in fresh SF900 III medium and 10 μL of each dilution was added in 12 replicates to the 96-well plate containing cells. Plates were incubated for 6–7 days at 27 ${}^{\circ }C$, after which wells were scored according to visualization of green fluorescence using a fluorescence microscope or cytopathic effects. Results were converted from TCID_50_ and reported as plaque forming units per ml (pfu/mL) [22].

### 2.3. Infections

Sf9 cells in suspension were infected with rBEVs at a density of ∼1.5–2 × $10^{6}$ cells/mL viable cells/mL at a MOI of ∼3 pfu/cell. Samples were harvested at the required times (hours post infection; hpi) wherein cells were centrifuged at 300× *g* for 10 min and resuspended in 2% paraformaldehyde diluted in phosphate buffered saline (PBS) for ∼30 min prior to analysis by flow cytometry where appropriate. The cell culture supernatant was kept at 4 ${}^{\circ }C$ for further analysis where appropriate.

### 2.4. Flow Cytometry and Analysis

Cells infected with GFP-producing rBEVs were analyzed using a FACSCalibur™ flow cytometer (BD Biosciences, San Jose, CA, USA) equipped with an argon-ion laser with an excitation frequency of 488 nm. Samples were run at the low flow setting (12 μL/min) and 10,000 events were collected. Analysis of flow cytometry data was performed using FlowJo^®^ V10 flow cytometry analysis software (FlowJo LLC, Ashland, OR, USA). Results are reported as the mean of at least 3 independent replicates.

### 2.5. SEAP Activity Assays

The supernatants of infections with SEAP-producing rBEVs were harvested by centrifugation at 1000× *g* for 10 min. SEAP activity was quantified using the SEAP Colorimetric Reporter Assay Kit (Novus Biologicals, Toronto, ON, Canada) according to manufacturer’s directions. The absorbance was measured using a Synergy 4 hybrid microplate reader (BioTek, Winooski, VT, USA) at a wavelength of 405 nm. The SEAP concentration was determined using a calibration curve of known SEAP concentration standards. At least 3 independent replicates were run for each rBEV and each sample was run in triplicate for quantification of SEAP.

### 2.6. Real-Time Reverse Transcription Polymerase Chain Reaction (RT-PCR)

RNA was extracted from infected cells using the Geneaid Total RNA Mini kit (FroggaBio, Concord, ON, Canada) and 500 ng was used as template for first-strand cDNA synthesis using the SensiFAST cDNA synthesis kit (FroggaBio) according to manufacturer’s directions. Real-time PCR was performed using the SensiFAST SYBR Hi-ROX kit (FroggaBio) according to manufacturer’s directions on an Applied Biosystems StepOnePlus™ Real-Time PCR System (Fisher Scientific). Primer pairs used for qPCR are given in Appendix A. Analysis was conducted in the R programming environment using the *pcr* package [23] using 28S rRNA as the internal reference gene for data normalization [24].

### 2.7. Bioinformatics

All bioinformatics analyses were conducted using the R programming environment using several Bioconductor packages, including *msa*, *genbankr*, and *ggbio* [25,26,27].

## 3. Results

### 3.1. Selection of *Ac*MNPV Promoters

The transcriptome data mapped to *Ac*MNPV open reading frames (ORFs) [18] as reads per kilobase of transcript per million mapped reads (RPKM) values were used to select *Ac*MNPV promoters with different expression characteristics. The ORFs were first divided into ‘classes’ based on the RPKM values (Table 2 and Appendix A) to separate them into groups according to relative transcript abundance. Selection of individual promoters for evaluation was based on two main criteria: transcription should accelerate and reach steady-state levels early in the infection cycle and maintain expression levels until the later stages of infection. The promoters from ORFs having a variety of steady-state RPKM values were selected for evaluation (Figure 1 and Table 1). The vast majority of commercially available BEVS transfer plasmids include either the polh or p10 promoter, while the gp64 and *hr5*-ie1-p10 promoters are present in two (Appendix A). As the difference in expression levels between the gp64/ie1 and polh/p10 promoters is extremely large (Figure 1A), the promoters selected for further analysis were expected to provide intermediary levels of expression between these levels (Figure 1B).

### 3.2. Evaluating the Expression Profile of *Ac*MNPV Promoters

Recombinant BEVs were prepared to express the model cytoplasmic protein GFP and secreted protein SEAP to assess the expression characteristics of the selected promoters. In addition to analysis of protein production, RNA was extracted from infected cells at 24, 48, and 72 hpi, and transcription of *gfp* or *seap* was analyzed using RT-PCR. Notably, for intracellular GFP production, the 39k promoter appeared to be the most active promoter during the earliest stages of the infection, with median fluorescence intensity of ∼100 au by 12 hpi, whereas the GFP levels were less than 10 au for the other promoters. By 24 hpi, the p6.9 promoter had produced slightly higher levels of GFP compared to 39k. The polh promoter, on the other hand, was among the lowest in terms of fluorescence intensity until 24 hpi, after which expression increased rapidly and by 48 hpi the polh-GFP rBEV produced the highest level of GFP. Production of GFP from the vp39 promoter was similar to the 39k rBEV in the later stages of the infection, however the ctx and orf75 promoter rBEVs were lower than expected from their respective RPKM values (Figure 2A and Table 3). The Δp10, 38k, and gp64 promoter rBEVs yielded fluorescence intensity measurements that were lowest of the promoters tested, with Δp10-GFP rBEV reaching ∼30 au by 48 hpi. RT-PCR data revealed that *gfp* transcript abundance from the p6.9 promoter was over two times greater than any other promoter at 24 hpi, and ∼7× higher than the polh promoter. By 48 hpi, transcription from the polh promoter was only slightly lower than the p6.9 promoter and at 72 hpi was nearly 3× greater than p6.9 (Figure 2B).

Expression and secretion of SEAP followed a similar trend to that of GFP. By 24 hpi, SEAP activity corresponding to ∼2–2.5 mg/L of SEAP protein had been secreted to the supernatant for the p6.9, 39k, and vp39 promoter rBEVs. Each of the other rBEVs had produced less than ∼0.5 mg/L SEAP (Figure 3A). By 48 hpi, SEAP activity had more than doubled for the p6.9-SEAP rBEV to ∼5.5 mg/L, whereas 39k and vp39 had produced ∼50% more (∼3–3.5 mg/L). The SEAP activity produced by the polh-SEAP rBEV, on the other hand, had increased by 5× in the same time interval and was similar to 39k and vp39. Despite a significant increase in transcription activity from the polh promoter between 48 and 72 hpi (and apparent sharp decrease in p6.9-*seap* transcription), SEAP activity reached ∼6.5 mg/L and ∼3.75 mg/L for p6.9-SEAP and polh-SEAP rBEVs at 72 hpi, respectively, (Figure 3A). Significantly, SEAP production from the 39k and vp39 promoters was not substantially different from that of the polh promoter despite significant differences in transcriptional activity (Figure 3B). Similar to the GFP results, the ctx and orf75 promoters produced lower SEAP activity than expected, while the other promoter rBEVs produced SEAP levels in a similar ranking to GFP expression; production of SEAP from the gp64 and 38k rBEVs were lowest but their order swapped as compared to GFP expression. Interestingly, RT-PCR results suggested lower *seap* mRNA abundance was produced from the ctx promoter than orf75 (Table 3).

### 3.3. Bioinformatics Exploration of Sequence and Genomic Architecture Determinants That Contribute to Promoter Activity

The promoter activity results prompted the exploration of specific sequence or genome architecture determinants that may impact promoter activity. Along with RPKM values for transcript abundance, putative transcription start sites (TSS) and promoter motifs were included in the transcriptome data set and were used for further analysis [18]. Additionally, sequences encompassing the 5${}^{\prime }$ untranslated region (5${}^{\prime }$UTR), TSS, and upstream sequences including the putative promoter motifs were extracted from the *Ac*MNPV genome and analyzed. Analysis was conducted using the ORFs divided into classes based on RPKM values. Unsurprisingly, the ‘Very Low’ class had the highest proportion of promoters with identified ‘TATA’ or ‘CAGT’ motifs, which indicate transcription from the host RNA Polymerase II (RNAP II). The ‘Low’ and ‘Medium’ classes had fewer RNAP II promoter motifs than ‘Very Low’, whereas the ‘High’ and ‘Very High’ classes had none (Appendix A). Similarly, the ‘TAAG’ motif, which is required for initiation of transcription from the viral RNAP (vRNAP) was present in the promoter regions for all ORFs in the Medium, High, and Very High classes, and was found less frequently in the promoters of ORFs in Low and Very Low classes. Interestingly, manual inspection of sequences around the TSS of several genes indicated deviation from the ‘CAGT’ motif. All of the examined ORFs with apparent deviations were designated as Low or Very Low transcript abundance. Inspection of the sequence surrounding the TAAG motif of several genes in each class found no deviation from this motif (data not shown). Next, the 5${}^{\prime }$UTRs were examined for differences between each class. The length of the 5${}^{\prime }$UTR (i.e., the number of nucleotides between the TSS and ATG start codon) was not statistically different between classes (Figure 4A). The A+T content, on the other hand, was significantly higher for the Very High class compared to the other classes (Figure 4B).

Various studies in the literature have suggested that the homologous repeat (*hr*) regions found in baculovirus genomes have roles as origins of DNA replication (*oris*) and as transcriptional enhancers that may act in both *cis* and *trans*. To investigate whether there was a relationship between transcript abundance and distance from a *hr* region, the genome map of *Ac*MNPV was colour-coded according to class (Figure 5) and the distance from the 5${}^{\prime }$ end of the *hrs* to the start codon of each gene was calculated (Figure 6). There did not appear to be any discernible relationship between ORF transcript abundance and proximity to any of the *hrs*.

A previous study identified two octamers (5${}^{\prime }$-ATTGCAAG-3${}^{\prime }$ and 5${}^{\prime }$-ATTAGGAA-3${}^{\prime }$ herein referred to as upstream and downstream, respectively) located within the sequences upstream of both the *p6.9* and *vp39* TSSs [28]. The authors hypothesized that these could be protein binding sites for late gene transcription. To test this hypothesis, sequences upstream of each *Ac*MNPV ORF (225 nucleotides upstream of the TAAG motif) were searched using these octamer sequences as queries. Only two matching sequences (upstream of the *p6.9* and *vp39* TSSs) were found for both queries. A search of the entire *Ac*MNPV genome only yielded 1 additional match for the downstream motif, which was located within the coding sequence of the *p74* gene, and 4 for the upstream motif, also found within coding sequences and not close to the TAAG motif for the adjacent downstream gene (data not shown). Two mismatches were required for both octamers to yield possible matches in the ∼225 nucleotides upstream of the TAAG motif for the majority of *Ac*MNPV ORFs, with multiple putative matches found for each (Figure 7). This same search was performed using the entire *Ac*MNPV genome sequence: 1733 and 1439 matches were found for the upstream and downstream sequences, respectively, and were roughly divided equally between the coding and noncoding strands for both. Allowing for 2 mismatches in the upstream octamer yielded 52 potential matches in the 29 individual sequences belonging to ORFs in the Very High, High, and Medium classes. Only 7% (2 of 29) did not contain a putative match, whereas 38% and 31% had 1 or 2 matches. Additionally, 11 different octamer sequences were present in more than 1 upstream sequence. Although the sequences were dispersed throughout the length of the upstream regions (Figure 7), a multiple sequence alignment (MSA) revealed that the consensus sequence was 5${}^{\prime }$-ATTGCAAN-3${}^{\prime }$. For the downstream octamer, 47 potential matches were found; however, 34% of the sequences had no putative matches, and only 31% had either 1 or 2. Inspection revealed that 8 sequences were found in at least two upstream regions (Appendix A). Similar to the upstream motif, although the consensus sequence was 5${}^{\prime }$-ATTAGGAA-3${}^{\prime }$, these sequences were located randomly throughout the putative promoter regions. Manual inspection of the upstream regions that only had a single putative match for either the upstream or downstream motifs revealed similar results.

Finally, sequences surrounding the late gene TSS and translation initiation site (TIS) were extracted for each ORF and analyzed. Multiple sequence alignments were created for each class and the consensus sequence for each MSA was calculated for comparison. The consensus sequences for the nucleotides flanking the translation initiation site (TIS) and transcription initiation site (TSS) are given in Figure 8A,B, respectively. Additionally, the consensus sequence for the most abundantly transcribed ORFs excluding *polh* and *p10* was also included for the sequences surrounding the TSS. Conserved nucleotides are highlighted in green and the most conserved locations have a box placed around them. Nucleotide positions −2, −3, and −7 with respect to the TIS (A is +1) appear to be the most conserved sites flanking the TIS, whereas −4, +6, +9, +10, and +12 (+1 is the first A in the TAAG motif) are highly conserved sites flanking the TSS.

## 4. Discussion

Since the first recombinant proteins were produced in the early 1980s, considerable research effort has been devoted to improving and optimizing expression of foreign genes in the BEVS [4]. The majority of this focus has revolved around increasing expression of foreign genes over that achieved with the endogenous polh or p10 promoters. This goal has led to studies identifying novel native promoters, chimeric tandem promoters, and introducing additional regulatory elements in *cis* or in *trans* [5,13,14,15,16], while this approach has yielded regulatory elements with higher expression profiles compared to polh, many of these reports have used the model cytoplasmic protein GFP for evaluation. Contrarily, several studies have suggested that weaker promoters than polh may improve production of proteins that require extensive post-translational processing and secretion [5,6,7,8]. Moreover, some multi-protein complexes such as VLPs have strict stoichiometric requirements for proper assembly, and overproduction of proteins may lead to inefficiencies and wasteful accumulation of unassembled proteins and/or formation of incomplete or improperly assembled particles. These deficiencies may impact efficacy of the biologic or economic feasibility of the production process [9]. To improve production of some recombinant proteins and biologics in the BEVS, it may be necessary to identify additional promoters with transcription characteristics that allow for the most efficient expression of each molecule. To augment the current catalogue of promoters for the BEVS, previously published transcriptome data [18] was used to identify promoters with transcriptional activities incrementally lower than the polh promoter, while several of the promoters have been identified previously, to our knowledge at least 3 have not been previously evaluated. Nevertheless, none of those selected have been compared in any systematic way nor routinely used for recombinant protein production.

To evaluate and compare the promoters, rBEVs were prepared to express the *gfp* and *seap* genes. For both reporter proteins, expression levels were largely consistent with the expected ranking compared to the RPKM values for each ORF. However, production of GFP and SEAP from the ctx and orf75 promoters were lower than expected, while expression from the 39k promoter was higher, particularly during the early stages of infection. This latter result agrees with a previous report in which *seap* mRNA transcribed from the 39k and p6.9 promoters were similar at 24 hpi [7]. Additionally, previous reports of novel promoters for the BEVS have shown that expression strength depended on the upstream sequence selected as the promoter; for example, a 120 bp sequence upstream of the *orf46* (pSeL120) gene of *Spodoptera exigua* MNPV (SeMNPV) produced nearly 2× more GFP fluorescence intensity in Sf21 cells compared to promoter sequences that were extended to either 140 bp (pSeL140) or 301 bp (pSeL). Deletion of the 25 nucleotides adjacent to the translation initiation site from pSeL, on the other hand, nearly abolished GFP production entirely [15]. Further, GFP expression from each promoter varied in different cell lines. In Se301 cells, pSeL140 and pSeL120 produced similar levels of GFP, while pSeL produced less than half the fluorescence intensity at 96 hpi. In Sf21 and Hi5 cells, however, pSeL120 remained high whereas pSeL and pSeL140 produced similar but lower levels of GFP. Interestingly, GFP produced from the *Ac*MNPV polh and p131 promoters, which is the *Ac*MNPV homolog of the SeMNPV *orf46* gene, produced similar levels of GFP in all 3 cell lines tested [15]. Given that the transcriptome data used for analysis here originated from *Ac*MNPV infection of High Five cells and baculovirus protein expression profiles may differ between infection hosts [18,29], the divergence between the expected and observed transcriptional strengths of the 39k, orf75, and ctx promoters in this report may be due to host specific factors or promoter sequence selection. Similar to the SeMNPV orf46 promoter, careful scrutiny of the specific sequences included may improve their transcriptional strength. Additionally, it is worth noting that the *ctx* gene is part of a polycistronic transcript with the *ac-bro* gene, which may have an impact on its transcription [30].

Unsurprisingly, expression of GFP with the polh promoter increased sharply at 24 hpi and was nearly 2× higher than with the p6.9 promoter by 48 hpi. Despite a drastic increase in transcriptional activity between 48 and 72 hpi, median GFP fluorescence intensity increased only ∼25%. However, by 72 hpi each of the infected cell populations had experienced precipitous declines in viability (data not shown). Increased permeability of intact dead/dying cells leading to leakage of GFP molecules out of the cell may account for the discrepancy between median fluorescence intensity and transcript abundance at 72 hpi, as has been observed previously [31]. Nevertheless, production of GFP from the other promoter rBEVs reached peak median fluorescence intensity by ∼36 hpi and maintained this level until 60 hpi after which fluorescence intensity declined, consistent with the decline in cell viability and results from the p6.9 and polh rBEVs. Interestingly, fluorescence intensity measurements for the 39k and vp39 rBEVs reached very similar levels, however production of GFP from the 39k promoter increased sharply by 12 hpi and reached peak levels by 24 hpi. The Δp10 promoter, described previously [32], is a truncated *Ac*MNPV p10 promoter in which the A/T-rich burst sequence for Vlf-1 binding between nucleotides +39 and +72 (relative to the +1 of the TAAG TSS) has been removed. Removal of the burst sequence from either the polh or p10 promoter regions strongly attenuates their transcriptional strength, however they initiate transcription at the same time and level as the full length promoters [33,34]. Consistent with these observations, in this study the Δp10 and polh promoters had similar fluorescence intensity values until 24 hpi after which the polh promoter’s transcriptional activity sharply increased whereas the Δp10 promoter increased GFP production very gradually to reach a maximum fluorescence intensity that was ∼2.5% that of polh at 48 hpi.

Expression of SEAP, on the other hand, was substantially higher from the p6.9 promoter than polh. Once again, despite a drastic increase in transcriptional activity between 48 and 72 hpi, SEAP activity from cell culture supernatants only increased by ∼25% to ∼3.5 mg/L for polh. This yield was only ∼50% that of the p6.9 promoter rBEV, and was statistically indistinguishable from the yield from either the vp39 or 39k rBEVs. This latter result supports an earlier study in which SEAP activity from transformed insect cells was substantially higher when the *seap* gene was transcribed from the 39k promoter than polh. Important differences in that study may have had an effect on those results; in addition to transformed cell lines, each promoter was linked to the *hr5* ori, which acts as a transactivator of several early genes including *39k*, but may be detrimental for polh activity [7,35,36]. In that study, despite transcription of *seap* being highest with the polh promoter, a significant proportion of the SEAP protein produced was found in intracellular protein extracts, indicating that inefficient secretion may have played a large role in the lower activity observed. Significantly, a recently published comparative transcriptome analysis of baculovirus-infected cells revealed significant differences in gene expression between rBEVs expressing model intracellular (mCherry) and secreted (Hemagluttinin; HA) protein products [37]. Notably, although the proportion of mapped reads were significantly lower for mCherry transcripts compared to HA transcripts, western blot analysis indicated that more mCherry protein was produced in both the intracellular and extracellular fractions as compared to HA. Several host cell genes were regulated specifically in response to the expression of secreted HA protein; many of the differentially regulated genes were involved in the stress response to unfolded or misfolded proteins, providing further evidence that protein folding and processing in the endoplasmic reticulum or Golgi apparatus is impaired or at its capacity when the polh promoter is employed [37].

Based on these results, further analysis was conducted aimed at identifying any genome architecture or sequence determinants that may impact transcriptional strength. The 5${}^{\prime }$UTR of the *polh* and *p10* genes are A/T-rich [33,38], and previous studies have reported improved foreign gene expression by inserting A/T-rich leader sequences in the 5${}^{\prime }$UTR of the polh promoter [39,40]. Additionally, productivity improvements have been reported by inserting the *hr1* or *hr5* homologous regions upstream of various promoters to enhance foreign gene expression in a *cis*-dependent manner [16,36,41,42,43,44]. However, these discoveries often require extensive experimentation and their effectiveness can vary widely depending on the promoters evaluated. For example, the effectiveness of including a 21 nucleotide sequence derived from the 5${}^{\prime }$UTR of a lobster *tropomyosin* cDNA sequence was based on extensive experience with expressing several variants of lobster Tropomyosin proteins [39]. Insertion of the *hr1* sequence upstream of promoter regions, on the other hand, significantly improved GFP production from the p10 promoter, but had no effect on either the p6.9 or polh promoters [16]. Interestingly, an earlier report suggested that inserting the *hr1* region downstream and in the reverse orientation in the *polyhedrin* locus contributed to hyperexpression of the foreign gene [43]. Another study reported that the *hr5* sequence strongly enhanced the 39k promoter but significantly impaired expression from the polh promoter [36]. We reasoned that if homologous regions could enhance transcription from any *Ac*MNPV gene, the location and orientation of the ORF with respect to the nearest *hr* may provide insight toward chimeric promoter design. Additionally, sequences surrounding the late gene promoter motif, upstream region, and sequences adjacent to the translation initiation ATG codon, as well as the entire 5${}^{\prime }$UTR, were analyzed for characteristics that may determine expression levels of each class.

Initially, the sequence of the 5${}^{\prime }$UTR for each *Ac*MNPV ORF was extracted from the *Ac*MNPV genome for further analysis of any differences that may be distinguishable between the aforementioned expression classes. The length of the 5${}^{\prime }$UTR (ie., the number of nucleotides between the TAAG motif and the ATG initiation codon) was not statistically different between classes, indicating that the length of 5${}^{\prime }$UTR sequence does not directly impact expression levels of *Ac*MNPV genes. The A/T content of the 5${}^{\prime }$UTRs, however, was significantly higher for the Very High class compared to the other classes. This is consistent with previous analyses of the *p10* and *polyhedrin* 5${}^{\prime }$UTR sequences [38,45], however in addition to the polh and p10 5${}^{\prime }$UTRs, the Very High class includes the *p6.9*, *odv-e18* and *odv-ec27* genes as well. Given that the p6.9 promoter was extremely effective for both intracellular and extracellular protein production, evaluation of the odv-e18 and odv-ec27 promoters may be pertinent. On the other hand, the A/T content between the Low and Medium classes were also significantly different, however the High class was not. It could reasonably be expected that if A/T content of the 5${}^{\prime }$UTR played a major role in gene expression, the High class would have higher A/T content than the Low and Very Low classes. Similar to the overall length of the 5${}^{\prime }$UTR, it appears that A/T content may also not be a significant determining factor in gene expression levels.

Next, the *Ac*MNPV genome was annotated according to the expression class each ORF was classified as and the distance in nucleotides was calculated between the 5${}^{\prime }$ end of each *hr* and the 5${}^{\prime }$ end of each *Ac*MNPV ORF. Cursory inspection of the colour coded genome map suggested that the most abundant ORFs were generally found in close proximity to homologous regions, however no discernible patterns were identifiable for any class. For example, the *p10* ORF, which was found to be enhanced by the insertion of the *hr1* sequence upstream of the promoter, is located ∼15 kilobases (kbp) from *hr1* in the *p10-hr1* orientation. The polh promoter, which was not influenced by *hr1* in the same orientation but contributed to hyperexpression of a foreign gene when it was placed downstream and in the reverse orientation of the expression cassette, is located ∼4 kbp from the *hr1* region in the *hr1-polh* orientation. Similarly, the *39k* ORF is located ∼46 kbp from *hr5*, which strongly enhances its activity. Although the *hr* regions can clearly enhance transcription of several promoters, there do not appear to be any obvious clues as to which specific promoters they may stimulate based on their genomic location or orientation.

The previously reported upstream and downstream octamer sequences were hypothesized to have a role in regulation of late gene expression [28]. The study noted that their spacing was similar in both sequence contexts; the upstream octamer was located 201 and 190 nucleotides upstream of the ATG initiation codon of the *vp39* and *p6.9* ORFs, respectively, whereas the downstream octamer was located 120 and 137 nucleotides from the initiation codons. The placement of these sequences with respect to the TSS was also similar: the 5${}^{\prime }$ end of the upstream sequence is 143 and 148 nucleotides from the TAAG TSS motif (nearest to the initiation codon) for *vp39* and *p6.9* ORFs, respectively. It was reasoned that if these sequences were important for late transcription, they would be enriched at positions adjacent to TSSs in the genome, and the upstream regions of highly expressed ORFs would contain octamers with more optimal sequences and positions, whereas they may be sub-optimal for less expressed ORFs, while allowing for 2 nucleotide mismatches yielded putative matches in the majority of these sequences, the random positioning of the octamers did not give us confidence that these may be sequences important for gene expression. Indeed, a genome wide search indicated that the upstream and downstream octamers with 2 mismatches appear in the genome (on either strand) at a rate of ∼13 and 11 times per 1 kbp, respectively, or ∼1.6 and ∼1.3 per 250 bp on one DNA strand. These rates are nearly identical to the average number of matches for each octamer in the 225 nucleotide upstream sequences. Given this result and the random dispersion of their locations, it is unlikely that these octamers are late gene transcription regulatory elements.

Relatively little is understood about the sequence factors that govern translation initiation (TIS) of baculovirus mRNAs. Although several studies have demonstrated that the mammalian consensus TIS (i.e., Kozak leader sequence) allows translation in the BEVs, the sequences flanking the most highly expressed *Ac*MNPV genes differ significantly from the Kozak sequence [19], while inserting the consensus Kozak leader sequence in the polh promoter did not improve expression of the human basic fibroblast growth factor, inclusion of bacterial, invertebrate, and A/T-rich synthetic leader sequences have resulted in substantial improvements in recombinant protein production [39,40,46,47]. Interestingly, similar to the polh promoter, the L21 sequence contains an A-rich stretch 9 nucleotides upstream of the initiation codon. This stretch is followed by sequence that is virtually identical to the consensus Kozak sequence, potentially indicating that the Kozak sequence may influence translation initiation of some foreign genes. Aside from this similarity, however, these sequences have relatively few nucleotides in common [39]. Further, a previous study systematically introduced all possible single-nucleotide substitutions in the nucleotides flanking the initiation codon of the *gp64* gene and found that substitutions at only 2 positions within the *gp64* ORF (positions +4 and +5) significantly impacted translation efficiency. The authors noted, however, that more complex relationships involving multiple nucleotide positions may have larger, additive, effects on translation initiation [48]. Although gene expression is an inherently stochastic process and transcription and translation are independent and discrete events [49], we reasoned that genes that are transcribed less may also have less efficient translation initiation. Accordingly, sequences flanking the ATG initiation codon were extracted from each *Ac*MNPV ORF, and consensus sequences were derived for MSAs from each expression class. Interestingly, although analysis of the *gp64* TIS suggested that nucleotides within the leader sequence upstream of the ATG had little impact on translation initiation, nucleotides at positions −2, −3, and −7 showed significant conservation, indicating they could be important for translation initiation. Although no clear patterns between classes emerged, there are differences that may be worthy of further exploration. For example, position −4 is a highly conserved A or less conserved W (A or T) for the Very High and High classes, respectively. The G to A substitution at the same position increased translation of GP64 by ∼10% [48]. Similarly, substitution of G and T with any other nucleotide at positions +4 and +5 within the *gp64* ORF, respectively, increased expression by ∼1.3–2.8 fold, while these positions within the *p6.9* ORF are 5${}^{\prime }$-GT-3${}^{\prime }$, only the *ac-bro* ORF has G at position +4 and no other sequence in the Very High and High classes are occupied by T at position +5, while these observations could be coincidental due to the small number of ORFs in the Very High and High classes, they could be worthy of further experimental scrutiny by introducing these mutations in single substitutions and in combinations to measure their effect on foreign gene expression.

Finally, determining promoter motifs and the underlining mechanisms that control gene transcription is a major goal of computational biology [50]. However, aside from the late gene TSS motif (5${}^{\prime }$-TAAG-3${}^{\prime }$), regulation of late gene expression of *Ac*MNPV is not well understood [18]. Aside from the previously described octamer sequences found upstream of the *vp39* and *p6.9* ORFs, a few studies have identified nucleotide regions that have a large impact on transcription by using linker-scan mutation and deletion strategies to systematically introduce targeted mutations and truncations within the promoter region of *Ac*MNPV promoters [28,34,38,45,51,52,53,54,55]. Many of these studies have suggested that only ∼15–20 nucleotides upstream and downstream of the TSS motif is required for full strength promoter activity, except for the polh and p10 5${}^{\prime }$UTRs which have binding sites for VLF-1 and are required for full activity [56]. However, each of these studies introduced multiple mutations simultaneously and among adjacent nucleotides, potentially obscuring complex and synergistic relationships between different nucleotide positions. For example, mutation of 7 or 13 nucleotides in the regions 9 to 18 nucleotides upstream or 6 to 19 nucleotides downstream, respectively, of the TSS of the *vp39* promoter resulted in ∼50% reduction in transcriptional activity, while mutating 9/10 nucleotides between positions −2 and −11 reduced expression to ∼10% compared to the control [34]. In addition to consensus sequences derived from MSAs from each expression class, a MSA and subsequent consensus sequence was calculated for a curated group of promoter sequences that included all of the promoter regions from the High and Very High classes with the exception of the 2 very late promoters, p10 and polh. Previous studies have suggested that the sequences surrounding the TAAG motif of these promoters have a lower affinity for the vRNAP and function as inefficient late gene promoters [34,56]. Similar to the analysis of flanking TIS sequences, few clear patterns emerged; however, some similarities may be worthy of further exploration. For example, position +7 is A in every sequence in the Curated class, and is T for polh and p10 and W (A or T) for the other classes. Similarly, −11 and −12 are conserved 5${}^{\prime }$-AA-3${}^{\prime }$ dinucleotide in the Curated class, however they are less conserved as promoter strength decreases. Interestingly, mutations that overlapped both of these sequences resulted in at least 50% reduction of vp39 promoter activity [34], and while these observations may be coincidental due to the small number of highly expressed *Ac*MNPV ORFs, they may be worthy of experimental scrutiny to further examine their importance to late gene expression regulation.

## 5. Concluding Remarks

In this study, promoters with transcriptional activity lower than the polh promoter were identified, characterized, and compared by evaluating expression of model cytoplasmic and secreted proteins. Although the polh promoter yielded the highest abundance of GFP, the p6.9 promoter produced nearly twice the amount of SEAP than polh, and the vp39 and 39k promoters yielded similar levels as polh. This adds further confirmation to previous reports in which weaker but earlier promoters resulted in higher yield and/or quality of recombinant proteins than the polh promoter, particularly those that require extensive post-translational processing and secretion. It is expected that the addition of these new promoters to BEVS arsenal may be useful for optimizing co-expression of individual protein constituents of complex biologics such as VLPs. Additionally, we used available transcriptomics and genomics data to scrutinize several determinants that have been previously hypothesized or suggested to be involved in late gene transcription regulation. As high quality transcriptome and proteomic data becomes more available, this general workflow may be useful in elucidating sequence determinants governing late gene expression to optimize promoter and baculovirus genome design.

## Figures and Tables

**Figure 1 viruses-14-02670-f001:**
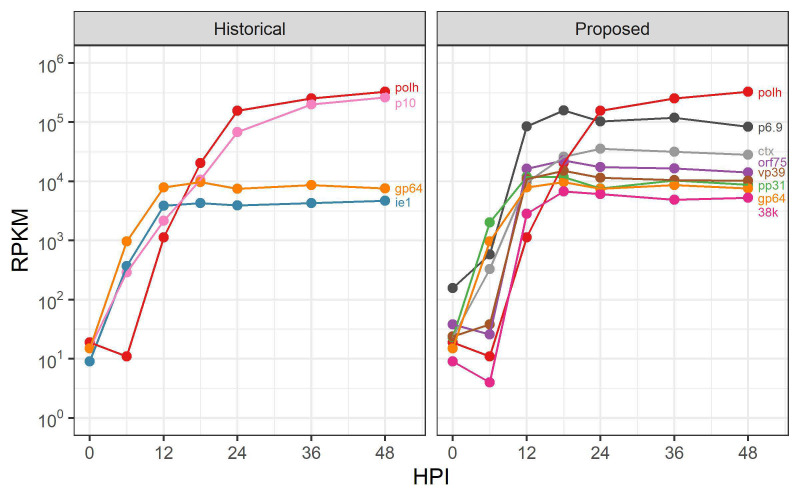
The promoters included on commercially available transfer plasmids have drastically different transcription profiles. (**Left panel**). The transcript abundance of *Ac*MNPV polh, p10, gp64, and ie1 ORFs, which are among the only promoters available on commercial transfer plasmids for foreign gene expression. (**Right panel**). The transcript abundance profiles of *Ac*MNPV ORFs selected for evaluation of the upstream promoter regions in this study. Promoters were selected for expression profiles between polh/p10 and gp64/ie1.

**Figure 2 viruses-14-02670-f002:**
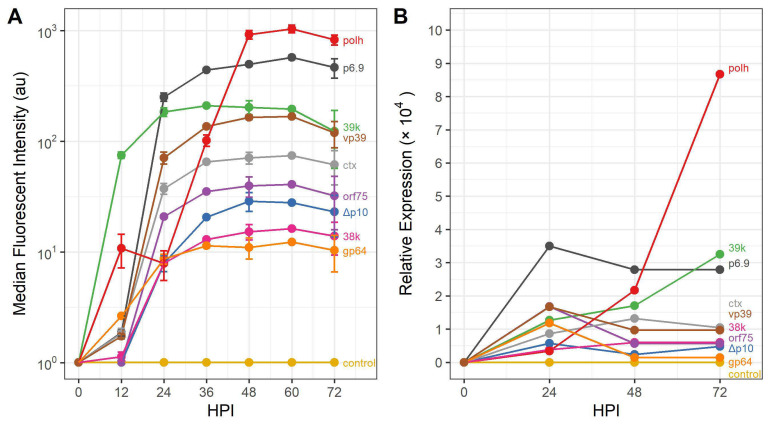
Production of intracellular GFP from selected *Ac*MNPV promoters. (**A**) Median fluorescence intensity measured using flow cytometry and (**B**) relative transcript abundance measured using RT-qPCR at various times post infection.

**Figure 3 viruses-14-02670-f003:**
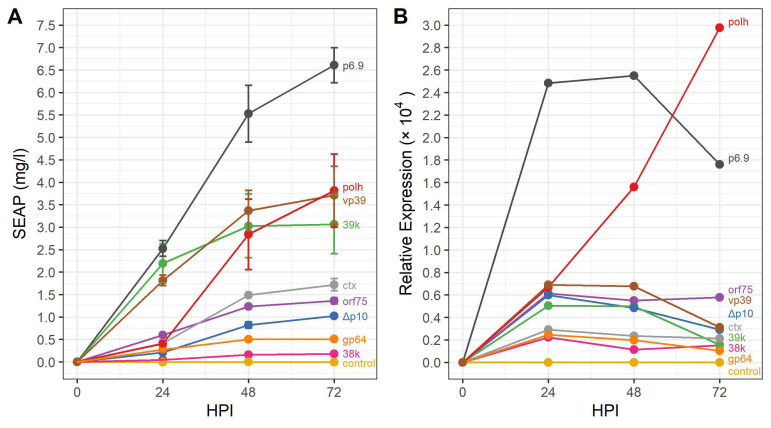
Production of extracellular SEAP from selected *Ac*MNPV promoters. (**A**) Yield of SEAP (mg/L) of culture supernatants measured using a colorimetric SEAP activity assay and (**B**) relative transcript abundance measured using RT-qPCR at 24, 48, and 72 h post infection.

**Figure 4 viruses-14-02670-f004:**
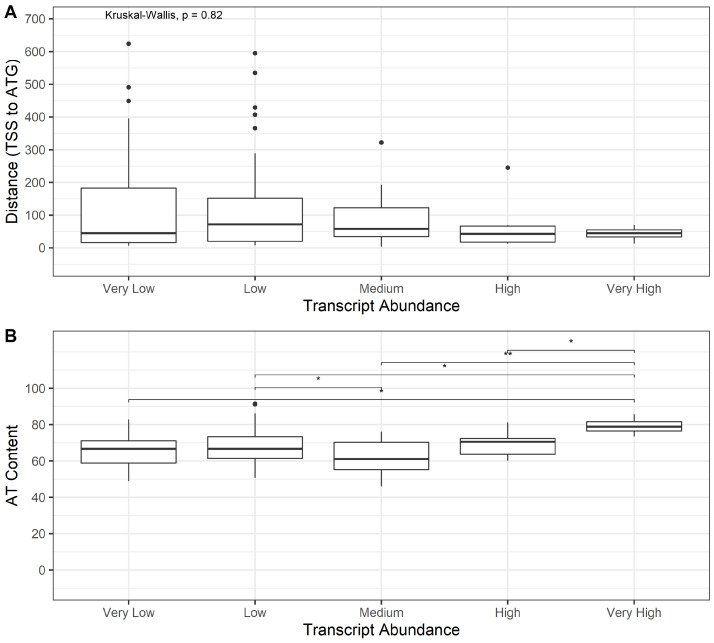
Evaluation of the 5${}^{\prime }$UTRs of *Ac*MNPV ORFs categorized according to transcript abundance. (**A**) Length (in nucleotides) and (**B**) A/T content of the 5${}^{\prime }$UTR between the late gene promoter motif (5${}^{\prime }$-TAAG-3${}^{\prime }$) and translation initiation codon (ATG). * *p* < 0.05, ** *p* < 0.01.

**Figure 5 viruses-14-02670-f005:**
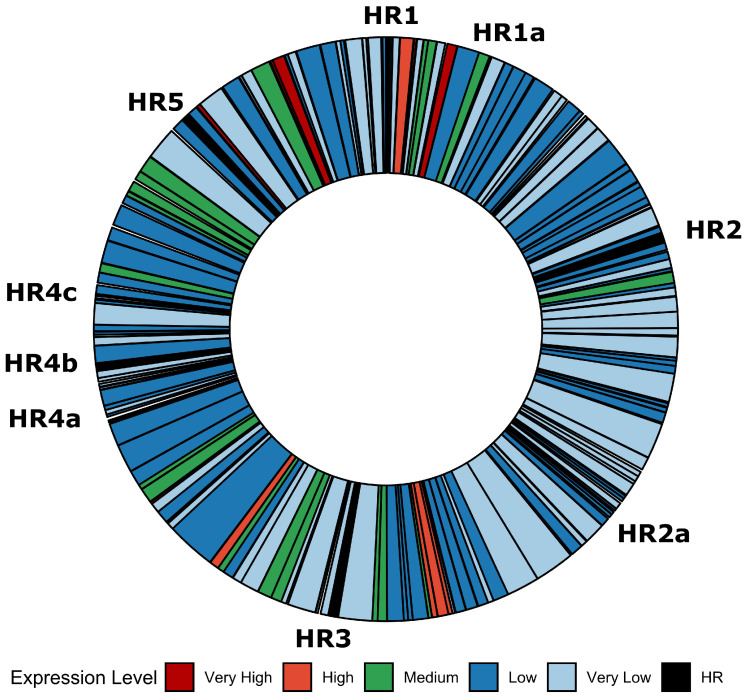
Circular chromosome map of *Ac*MNPV ORFs colour-coded according to transcript abundance.

**Figure 6 viruses-14-02670-f006:**
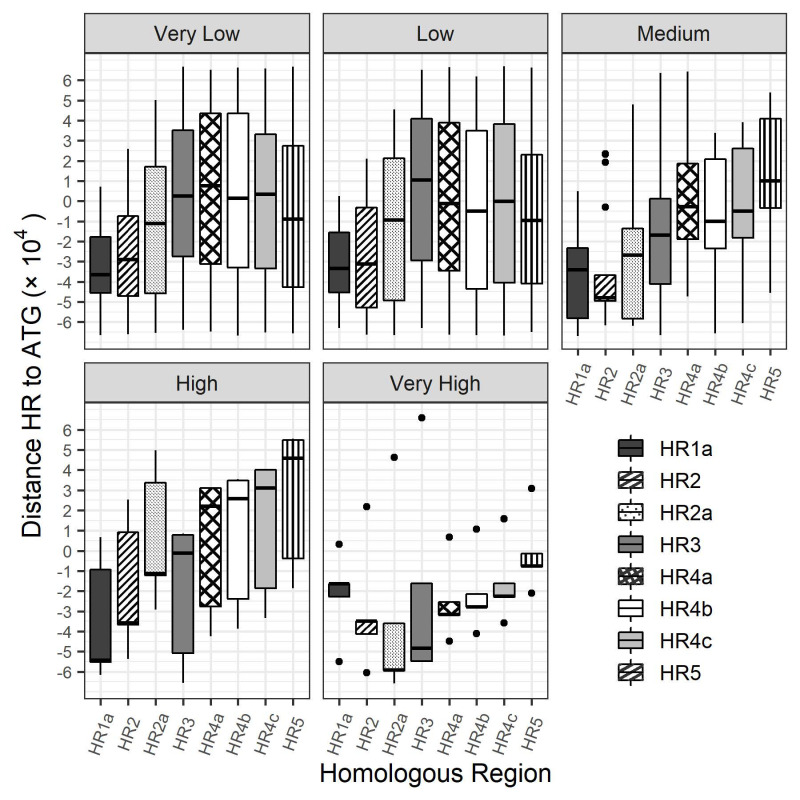
Distance (in nucleotides) between the start codon and 5${}^{\prime }$ end of each homologous region for every *Ac*MNPV ORF categorized according to transcript abundance. The distance was calculated by subtracting the genomic location of the 5${}^{\prime }$ end of each *Ac*MNPV ORF from the genomic location of the 5${}^{\prime }$ end of each *hr*. Positive values represent ORFs located behind (clockwise) to the *hr* and negative values represent distances between ORFs that are located in front of (counterclockwise) the *hr*.

**Figure 7 viruses-14-02670-f007:**
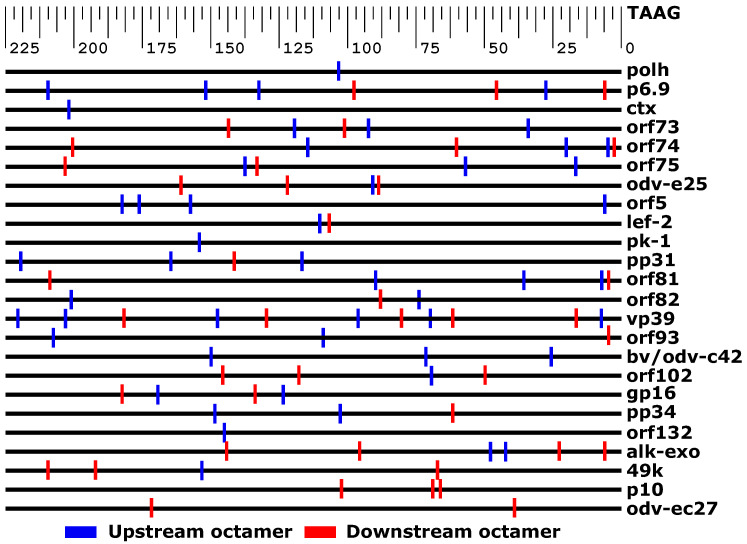
The approximate positions of the upstream octamer (blue) and downstream octamer (red) in regions upstream of the late gene promoter motif for the most abundant *Ac*MNPV ORFs.

**Figure 8 viruses-14-02670-f008:**
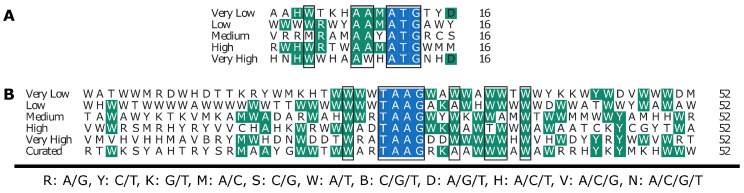
Consensus sequences calculated from multiple sequence alignments. (**A**) Consensus sequence for the nucleotide sequences flanking the translation initiation site and (**B**) the late gene promoter motif. Consensus sequences were calculated from multiple sequence alignments of sequences extracted from *Ac*MNPV ORFs that were categorized according to transcript abundance.

**Table 1 viruses-14-02670-t001:** AcMNPV genomic coordinates for promoters used in this study.

Promoter	Coordinates ^†^	Predicted Promoter Motifs ^‡^	Direction	Class
polh	4428…4519	TAAG	+	Very Late
p6.9	86,889…87,204	CAGT, TAAG	−	Late
ctx	2246…2447	TAAG	−	Late
orf75	63,528…63,912	TAAG	−	Late
vp39	76,578…77,103	TAAG	−	Late
39k/pp31	30,070…30,398	TATA, CAGT, TAAG	−	Delayed Early
gp64	109,718…110,022	TATA, CAGT, TAAG	−	Early/Late
38k	85,984…86,276	TAAG	−	Late
Δp10	118,635…118,808	TAAG	+	Very Late

†: based on NCBI ref. NC_001623.1 [19]; ‡: based on ref. [18].

**Table 2 viruses-14-02670-t002:** RPKM value ranges for *Ac*MNPV transcript ‘classes’.

Class	RPKM Range	ORFs
Very High	RPKM ≥ 50,000	5
High	20,000 ≥ RPKM < 50,000	6
Medium	10,000 ≥ RPKM < 20,000	17
Low	1000 ≥ RPKM < 10,000	65
Very Low	RPKM < 1000	56

**Table 3 viruses-14-02670-t003:** Promoters ranked by GFP and SEAP production at 48 hpi.

Promoter	Rank (48 hpi)				
	ORF	GFP		SEAP	
	RPKM	Protein	qPCR	Protein	qPCR
polh	1	1	2	2	2
p6.9	2	2	1	1	1
ctx	3	5	4	5	7
orf75	4	6	6	6	4
vp39	5	4	5	3	3
39k	6	3	3	4	5
gp64	7	9	9	8	8
38k	8	8	7	9	9
Δp10	n/a	7	8	7	6

## Data Availability

The datasets generated during and/or analysed during the current study are available from the corresponding author on reasonable request.

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
