# Peer review of "Utility of Alternative Promoters for Foreign Gene Expression Using the Baculovirus Expression Vector System"

_viruses, 2022, doi:10.3390/v14122670_

Round 1

Reviewer 1 Report

The introduction of the manuscript is vey well written explaining all the hardship faced on the BEVS in terms of selected promoter for expression of specific genes, especially the ones which needs to be co-expressed. This explains already why the work presented here is important to improve BEVS system in the near future. The way to determine better promoter to raise the expression with transcriptome data,  comparison analyses on GFP and SEAP genes expressions, and bioinformatics analyses on the sequence and genomic architecture determinants for promoter activity were nicely documented and would be a solution for low expression of specific genes via BEVS due to wrong promoter selection. Especially, bioinformatics analysis would bring a new point of view to the field so people would pay more attention to the sequences to increase the promoter activity. The only part that can need to be modified is that there is a big emphasis on the co-expression difficulties of subunits of multi protein complexes like VLPs due to stoichiometry that’s why from introduction part, I had an impression that you were about to check the expression levels of specific VLPs such as rotavirus-like particles with different promoters that you have identified from your transcriptome and bioinformatics analyses. I very much like the research that you have presented here but I recommend that the emphasis on the complex expression difficulties due to selected promoter can be decreased or if you have already tried to express multi protein complex with the promoters that you have found giving better expression outcomes, the data can be added to the manuscript.

Author Response

The authors thank the reviewers for their critique of the manuscript.

With respect to "The only part that can need to be modified is that there is a big emphasis on the co-expression difficulties of subunits of multi protein complexes like VLPs due to stoichiometry that’s why from introduction part, I had an impression that you were about to check the expression levels of specific VLPs such as rotavirus-like particles with different promoters that you have identified from your transcriptome and bioinformatics analyses. I very much like the research that you have presented here but I recommend that the emphasis on the complex expression difficulties due to selected promoter can be decreased or if you have already tried to express multi protein complex with the promoters that you have found giving better expression outcomes, the data can be added to the manuscript."

We do not have additional data to add to the manuscript at this time. This manuscript brings us one step closer to the control of expression, where the ultimate goal is to use these different promoters to gain better production of VLPs.

Reviewer 2 Report

The manuscript entitled " Utility of alternative promoters for foreign gene expression using the baculovirus expression vector system¨ is a study focused on the evaluation of alternative baculovirus promoters to direct the expression of recombinant protein. The authors make an in-deep characterization of genome architecture and sequence determinants that may impact transcriptional strength using bioinformatic tools. This is the firs report that effectively uses a bioinformatic approach to study and evaluate promoter sequence in baculovirus. It is a field that has not yet been sufficiently explored. The work represents a great advance in the study and improvement of the baculovirus expression system. This has opened the possibilities to add new promoters, alternative to the traditional ones such as polh, to baculovirus expression catalogue for different purpose. The work is well written and is of interest for the scientific community because offer new tools. The experiments are well designed.

I agree with the publication of the article in Viruses.

Minor revision:

Point 2.2. Line 107-108. How do you convert TCID50 to pfu/ml? it there any reference related with this formula?

Author Response

We thank the reviewers for the critique of our manuscript.

With respect to: Point 2.2. Line 107-108. How do you convert TCID50 to pfu/ml? it there any reference related with this formula?

This conversion factor is based on the Poisson distribution and has been utilized for many years. We have added the following reference to the manuscript.

Yang JP. Small-Scale Production of Recombinant Proteins Using the Baculovirus Expression Vector System. Methods Mol Biol. 2016;1350:225-39. doi: 10.1007/978-1-4939-3043-2_10. PMID: 26820860.